# Secure Firmware Update: Challenges and Solutions

Luigi Catuogno [1,*,†] and Clemente Galdi [2,†]

1. Department of Economics, Law, Computer Science and Motor Sciences, Università degli Studi di Napoli "Parthenope", 80133 Napoli, Italy
2. Informatica Presso il Dipartimento di Studi Politici e Sociali, Università degli Studi di Salerno, 84084 Fisciano, Italy; clgaldi@unisa.it
* Correspondence: luigi.catuogno@uniparthenope.it
† These authors contributed equally to this work.

**Abstract:** The pervasiveness of IoT and embedded devices allows the deployment of services that were unthinkable only few years ago. Such devices are typically small, run unattended, possibly on batteries and need to have a low cost of production. As all software systems, this type of devices need to be updated for different reasons, e.g., introducing new features, improving/correcting existing functionalities or fixing security flaws. At the same time, because of their low-complexity, standard software distribution platforms and techniques cannot be used to update the software. In this paper we review the current limitations posed to software distribution systems for embedded/IoT devices, consider challenges that the researchers in this area have been identifying and propose the corresponding solutions.

**Keywords:** firmware update; software update; IoT; embedded devices; mobile devices

## 1. Introduction

The increasing use of Internet connected devices, has improved people's everyday life. Such devices can be used as building blocks for structured services that can be easily deployed. They can be as tiny and simple as smart lamps, or have considerable computing and storage capacities as home security controllers.

Given the huge number of possible devices that are currently available on the market, the possibility that each one them runs multiple payloads and the corresponding wide range of possible compositions of provided services, heterogeneity becomes a crucial property that characterizes every IoT based system, independently from the specific application it has been deployed for.

One key element that makes IoT devices widespread is the possibility of creating or connecting to networks. At the same time, many devices have been designed to operate on batteries, imposing the strong limitation of reducing as much as possible the power needed to carry out all operations. Among all components of an IoT device, the one that consumes more energy is by far the one devoted to the communication. This makes the design of software components that have to be deployed on such devices particularly challenging since the tradeoff between information dissemination, needed to correctly run the prescribed task, and power consumption, necessary for guaranteeing device survivability, becomes part of the software lifecycle since its early stages.

On the hardware side, a crucial feature of IoT devices is their cost. Clearly, better performance/features increase the cost of each device and one important issue is to properly balance device costs and the desired device functionalities. In general, devices that provide some type of hardware-supported security primitive have higher cost but, also, corresponding higher security guarantees.

In order to be competitive, device manufacturers need to continuously release new products with short time-to-market and lowest possible prices. To this end, in this specific

market, it is extremely hard to produce devices with strong security guarantees. Furthermore, since new security threats are continuously discovered, each manufactures needs to put in place a methodology to identify threats that apply to its products, produce firmware/software updates that mitigate/solve the threat and securely update all the devices that are subject to the identified flaw.

Addressing the need of firmware/software update is by itself a very challenging task in this context for the following reasons:

- The number and features of stakeholders involved in different products deployments may vary significantly. Each deployment site may have its own update/upgrade policies that may depend on the context and/or on internal security/operational policies.
- In a typical deployment, multiple devices from multiple manufacturers need to share the same operational context. In many case, interoperability might be an issue and the uncoordinated updates of a subset of devices might lead to service unavailability. Furthermore, devices might even be unattended and prone to faults due to external conditions.
- Each manufacturer needs to provide a scalable software distribution system that is able to operate with thousands/millions of devices, in an asynchronous manner, while guaranteeing correct delivery and software integrity even when devices might not be able to execute complex security protocols.

These challenges have been largely addressed in different contexts, each characterised by a specific set of requirements. Each solution in the literature has been designed with a specific set of requirements, making it secure/efficient in the considered domain but, at the same time, possibly insecure/inefficient in other contexts. In this paper we will present in details requirements and restrictions that might appear in different operational contexts and discuss solutions to the firmware distribution problem that have been presented in the literature.

The rest of the paper is organized as follows. In Section 2, we report the reference scenarios that are usually found in the literature. In Section 3 we discuss methodologies and technologies that can be considered as building blocks for secure software update systems. In Section 4 we describe software distribution models and systems that have been presented in the literature. Finally, a discussion on open challenges will conclude the paper.

## 2. Reference Scenario

Managing the deployment of firmware and software updates for devices operating within a IoT ecosystem is a pretty delicate task as it has a considerable impact on the reliability and security of single devices as well as the whole infrastructure built upon them. Furthermore, IoT and embedded devices are, nowadays, the preferred target of plenty of attack vectors. This is due to several reasons that include:

Devices have, often, limited computational resources/capabilities so that, in some cases, such constrained devices are not suitable to fully implement security protocols, peers' authentication, data encryption algorithms and so on. The security of the whole system may depend on the size of cryptographic keys that, because of such limitations, are forced to be short. Limited bandwidth and/or high latency may be used to induce inconsistencies in the different nodes in the network.

Devices might be installed and operate within a logical/physical domain which spans over a wide area. This makes harder the design and setup of any centralized protection of the domain boundaries. Furthermore, many devices frequently move across such boundaries, e.g., all wearable devices follow their owner in her everyday activities. This creates the need of designing software update systems that are resilient to attacks coming from a maliciously programmed herd of transient nodes.

Devices operation might be mostly unattended and, thus, time might pass before any potential intrusion or fault is discovered. In particular, device updates are mostly accomplished in a fully automated way, hence, update procedures and protocols should

be able to promptly detect any update fault and possibly revert the procedure in order to bring the device back to a functional state.

IoT infrastructures may contain a noticeable number of (potentially heterogeneous) devices. Update procedures and protocols should be designed to scale as well. A critical aspect is that large scale updates take some time to be completed. In the meantime, devices equipped with different firmware versions might coexist for a while and, moreover, several faults might occur, making unavailable a certain number of devices. This may affect the security and the reliability of the whole infrastructure (of some parts of it) whenever, for example, devices running different firmware versions turn out to be partially or fully incompatible.

## 3. Enabling Methodologies and Technologies

In this section we present basic methodologies and technologies that can be used to build secure software distribution systems.

### 3.1. Hardware Classification

Embedded systems include a wide variety of equipment and computing devices (both singles or grouped into cooperating architectures) which range from mobile hand-held terminals, intended for general purposes, to highly specialized, variously sized and shaped devices such as controllers, peripherals, RFID Tags, sensors etc. The characteristics of such devices vary as well in terms of interaction kind and mean, purpose, computational capabilities and resources.

Major applications of Embedded Systems feature multiple embedded devices, each running its local software component, accomplishing its tasks and cooperating with the others. The software every component runs, can be seen as a single distributed architecture which features front-end applications, intermediate middleware layers and on-board OEM firmware along with communication links and protocols used for the sake of overall system coordination, interface and management.

In particular, IoT encompasses those embedded systems whose components communicate each others through TCP/IP based communication protocols and hence, potentially, through public segments of the Internet. Very often, IoT architectures entail the massive deployment of very cheap and constrained devices, with little (or none) interface, except the network adapter, and limited computing capability.

Ensuring security, integrity and availability of the overall system is a critical aspect in essentially all real-life applications that use IoT/embedded devices, e.g., [1–4]. Cryptographic algorithms and security protocol are, in general, resource demanding tasks. To this regard, design choices result in a trade-off between the security goals and the involved devices capabilities.

Such decisional process may benefit from a standard device classification. A good state-of-the-art can be found in [5] where the authors list a number of features that can characterize each device. Examples are the size of the bus, that typically corresponds to the CPU word size, the RAM size, the clock speed, the supported OS, the CPU power usage, the type of supported communication mean, whether or not the processor supports asymmetric cryptographic primitives. Given these features the authors identify 5 different classes of devices, ranging from class 1 devices, corresponding PC-like devices, with MBs/GBs RAMs and 64-bit processors, with high speed-wireless communication and running commercial operating systems, to class 5 ones in which the CPU word size can be as low as 4 bits, the RAM size ranges from few hundreds of byte to few kilobytes, slow processors, no asymmetric cryptographic support, and so forth.

Looking closer to the IoT world, RFC 7228 [6] defines a classification for constrained node networks consisting of three classes, where class-0 devices essentially correspond to sensor-like motes, class-1 includes devices that are able to execute specifically designed low-demanding protocol stacks and, finally, class-2 devices are capable of executing notebook-like protocols while still having limited memory and computational capabilities.

### 3.2. Trusted Execution Environments

Trusted Execution Environments (TEE) [7] consist of a set of hardware and software components which enable a computing device to set up an architecture which put side-by-side a Regular Execution Environment (REE) and one or multiple Trusted Execution Environments. REE runs legacy OSes and applications (Regular Applications or RAs) which are considered untrusted, while TEEs run a so called Trusted Applications (TA). TEEs ensure code authenticity, integrity and confidentiality for running TAs along with confidentiality of data any TA handles. Within a typical TEE-enabled platform, TAs are intended to act as trusted back-ends for RAs, accomplishing critical tasks such as handling cryptographic keys and user credentials.

Originally, once installed, TAs were intended to remain immutable. Alternatively, main tasks of TAs lifecycle used to be carried out within OS or firmware updates which were generally accomplished having the physical control of the device. Such a model, which does not take into account the wide variety of actors and roles that could be involved into TAs development and deployment, has been largely improved with the TEE Management Framework Specifications (TMF).

The TMF specifies a set of operations for TEEs administration and models the involved actors, their roles, relations and hierarchy through the concept of security domain. Moreover, TMF specifies a security layer for authentication and secure communication.

TMF's TA management operations, effectively fulfills the requirements of secure software distribution amongst TEE-powered target devices. However, despite GlobalPlatform compliant TEEs implementations are available for different hardware platforms including ARM TrustZone [8], Intel SGX [9] and AMD-SEV.

### 3.3. Remote Attestation

Remote Attestation [10] is a security service [11] through which a verifier is able to obtain a trustworthy measurement of the internal state of an untrusted device: the prover. The word "Remote" does mean that the attestation process takes place by means of a network protocol. Remote Attestation protocols essentially fall in two categories: Software- and Hardware-based.

In Software-based Remote Attestation, the verifier has an "intimate" and direct knowledge of some immutable characteristics of the prover (the attested device), e.g., the precise time it needs to carry out a certain operation and how it impacts on its registers as well as the capability to map its whole memory. So that, with the attestation process, the verifier first challenges the prover, validates its response and verifies that the response computation produced the expected side-effects (took the expected time) on the prover's state.

This technique fits well enough those scenarios in which verifier and prover are connected with no intermediates, whereas such kind of solutions have proved to be less effective if applied to networked devices.

In Hardware-based Remote Attestation, the prover leverages a secure hardware component to provide the verifier of a cryptographic proof of its state authenticity. The proof includes a digitally signed measurement of a certain prover system component such as checksums of in-memory code images or registers state. Cryptographic keys and algorithms used to compute the proof are stored in a tamper-resistant storage and are not under the control of the legacy software running on the device.

Plenty of solutions for Hardware-based attestation have been proposed. These are built upon on of the different off-the-shelf secure hardware components such as Trusted Computing's TPM [12], ARM TrustZone [8] and Intel SGX [9], that implement strong cryptographic primitives and protocols.

However, such solutions are not suitable for low-cost and computationally constrained devices that are poorly (or not at all) equipped with cryptographic hardware and that constitutes a significant "population" of any embedded/IoT ecosystem.

### 3.4. Lightweight Cryptography

Secure firmware/software distribution systems need to guarantee basic security properties, namely payload confidentiality, integrity, authenticity and availability. In many cases, software confidentiality needs to be protected as part of the IPR manufactures policies. Furthermore, a publicly available firmware/software can be used by an adversary to obtain information about its vulnerabilities that, in turn, can be used to mount an attack. Clearly software integrity is a prerequisite to provide a device that will be working properly after the software has been installed. Software authenticity guarantees the customer that the software that is going to be installed has been developed by a software manufactures that has been authorized by the product manufactures. Finally, software availability ensures that each product owner is able to update the software whenever deemed necessary and, at the same time, provides software manufactures a reliable way to timely distribute updates, e.g., whenever a security flaw is identified.

Cryptographic schemes are fundamental building blocks for guaranteeing, in a strong sense, security properties. Indeed, practically all softwares that use the Internet for implementing their functionalities, implement some kind of cryptographic primitive to enforce some security property.

The need of cryptographic primitives becomes even more important when considering IoT/mobile devices. Indeed, in these contexts, by their nature, the devices need to use to some extent networking in order to implement the intended functionalities.

Depending on the application context, the security of communications has to be intended properly. For example, in the context of IoT, in many cases the confidentiality of sensed data may not be an issue, e.g., when measuring the temperature in a given area. On the other hand, the authenticity and the integrity of such data guarantee that the decision process that uses them is not subject to any external malicious manipulation. From this point of view, end-to-end security has to be intended as "application-to-application" security, as opposed to "device-to-device" security. In many application scenarios, data might be collected on one (IoT-device-)end and pass through a number of intermediate devices that may pre-process/aggregate them before reaching the final processing end. Along this path, some connections might be already protected by some type of cryptographic primitive, e.g., data-link encryption in cellphones. Nevertheless, application-to-applicaton cryptography is desirable in order to ensure security regardless of the underlying communication system.

It is well known that cryptographic primitives implementations are resource demanding. Clearly, their adoption in the IoT/mobile settings has to consider strong constraints of these types of devices. Specifically, IoT/mobile devices are typically limited with respect to processing capabilities and power consumption. Nevertheless, lightweight cryptographic primitives need to guarantee high security standards.

Roughly speaking, cryptographic primitives can be partitioned into symmetric and asymmetric ones. Symmetric cryptography assumes the existence of a key that is shared between two parties and that they keep secret. Symmetric schemes are, in general, lightweight as they are designed to process few tens of bits at the time, the block, and typically, they can be efficiently implemented in hardware. These primitives can be effectively used when the set of parties is somehow static, small and known in advance.

Asymmetric cryptographic primitives use a pair of keys, known as the public key and the secret key, that are mathematically related in a way that, (informally) one key can be used to invert or verify the operations executed with the corresponding key in the pair. This class of primitives allow secure interaction when the set of agents is dynamic, e.g., when there is the need to communicate with an unknown device. This flexibility comes at the cost of speed. Asymmetric primitives are thousands of times slower than symmetric ones, require much more computations since they're based on mathematical operations over variables consisting of hundreds or even thousands of bits.

Security of cryptographic primitives depends on a number of factors, one of which is the key size. For secure primitives, the longer is the key, the more secure is the scheme. It is

important to stress that the security of primitives of different classes or based on different assumptions are not comparable simply by looking at their key size.

Literature on cryptographic research is vast and in continuous evolution. For the purpose of this paper we first focus on schemes and solutions that have been subject to a standardization process.

ISO/IEC 29192 [13] is a family of standards that focuses on lightweight cryptographic primitives "suitable for lightweight cryptographic applications, including radio-frequency identification (RFID) tags, smart cards (e.g. contactless applications), secure batteries, health-care systems (e.g. Body Area Networks), sensor networks, etc.". It currently consists of 8 parts each dealing with different types of primitives. Specifically, part 1 presents terms and definitions. ISO/IEC 29192 part 2 presents three lightweight block ciphers, PRESENT [14], 64-bit block size, 80 or 128 bit key size, CLEFIA [15] and LEA, both with 128-bit block size and 128, 192 or 256-bit key size while part 3 is devoted to the stream ciphers Enocoro [16,17] and Trivium [18]. Part 4 introduces mechanisms based on lightweight asymmetric cryptography. Specifically, cryptoGPS, an identification scheme, ALIKE, a mechanism for authentication and key exchange, and an identity based signature scheme. ISO/IEC 29192 part 5 defines three hash functions, PHOTON [19], SPONGENT [20] and Lesamnta-LW [21]. ISO/IEC 29192 part 5 defines three Message authentication codes, namely, LightMAC [22], Chaskey-12 [23] and "Tsudik's keymode". Part 7 deals with broadcast authentication protocols while, part 8 presents a method for authenticated encryption.

ISO/IEC 29167 [24] is a family of standards that presents solutions to secure RFID communications. These mechanisms can be used only with RFIDs air interfaces that have security mechanisms onboard. ISO/IEC 29167:1 presents a general framework for the development of security mechanisms. The other parts of the standard ISO/IEC 29167, presents security services based on block ciphers AES-128 (part 10), PRESENT-80 [14] (part 11), AES OFB (part 14), RAMON [25] (part 19), SIMON [26] (part 21) and SPECK [26] (part 22), ECC-DH key agreement (part 12), on the stream cipher Grain-128A (part 13), XOR (part 15), on Elliptic Curve cryptographic primitives ECDSA-ECDH (part 16), on the cryptoGPS identification scheme (part 17).

Elliptic Curve Cryptography has, in general, a fundamental role in the design of secure mechanisms and, in particular, when such mechanisms have to be executed on constrained devices. This is essentially due to the possibility of having short keys and efficient algorithms that can be implemented and executed on IoT/mobile devices, while keeping high security levels. Such importance is testified by a large number of standards that deal with this type of cryptosystems. ISO/IEC 15946 [27] family describes the mathematical background and the curve generation algorithms that are at the basis of primitives like ciphers ISO/IEC 18033-2 [28], digital signatures ISO/IEC 9796-3 [29], ISO/IEC 14888-3 [30] key management ISO/IEC 11770-3 [31] and others.

The new standard for lightweight cryptography selected by NIST at the end of the NIST Lightweight Cryptography competition (2019–2023) [32] is a family of authenticated encryption and hashing algorithms called Ascon [33]. Such family of primitives had already been selected as winner of the Ceaser competition (2014–2019) [34].

In recent years, the advances in the design and prototyping of quantum technologies, on one side, and on quantum computing, on the other side, have made clear the need of developing cryptographic primitives that were able to resist to threats posed by adversaries able to effectively use quantum computing technologies. In this direction, the NIST launched the Post-Quantum Cryptography Standardization [35], a project for the selection, evaluation and standardization of quantum-resistant public-key cryptographic algorithms. At the time of this writing, the process has reached its fourth round and the algorithms that are currently under evaluation are a public-key encryption and key-establishment algorithm named Crystals-Kyber, and three digital signature algorithms, namely, Crystals-Dilithium, Falcon and Sphincs+.

To this end, the field of secure software updates is not an exception. As observed by the IETF Software Updates for Internet of Things (SUIT) group in [36], since the lifespan of IoT

devices may last decades, manufacturers should start considering the use of post-quantum cryptographic primitives. There are few studies, e.g., [37,38] that analyze the feasibility and performance of quantum-resistant cryptographic primitives on IoT-class devices.

What we've discussed so far are cryptographic primitives that have been developed for IoT devices and that have undergone a standardization process. There are, of course, many other secure cryptographic primitives that can be deployed on IoT devices but that have not been standardized, yet. In general, ECC-based cryptographic primitives can be used to design Identity-based cryptographic schemes [39] in which the public key of an entity can be derived from its publicly available identity, e.g., a unique arbitrary identifier, an email address, etc. The use of ID-based cryptographic primitives reduces the need of a secure infrastructure that keeps the association between identities and their corresponding public keys. Whenever a public key for a specific identity is needed, everyone can compute it starting from the identity and the scheme public parameters. A generalization of ID-based schemes are the Attribute-Based Encryption (ABE) primitives [40], in which the the secret key of the entity is related to some attributes the entity may possess, e.g., a specific processor type or a specific amount of memory, etc. In the case of encryption, the ciphertext is dependent on a set of attributes and the decryption is possible only if the attributes of the decrypting entity match the decryption policy. There are essentially two families of ABE schemes. In Key policy ABE, (KP-ABE) [41], the ciphertext is associated with a set of attribute labels and the decryption policy is embedded in the key sent to the user. In this case, the key generator is able to dynamically redefine the access policy. In contrast, in Ciphertext-policy Scheme, (CP-ABE) [42], the decryption policy is embedded in the ciphertext and, thus, the encrypting entity (statically) defines the decryption policy associated to a specific message. Attribute-based cryptographic primitives constitute flexible tools for enforcing access control on encrypted data.

### 3.5. Intelligent Networking

A central role in IoT/mobile devices lifecycle is clearly played by the networks on which such devices are deployed and through which the firmware is delivered. By the end of 1990s, the increasing amount of data available on the Internet and complexity of applications made clear the need of an evolution in the support for the development, deployment and operation of Internet services. The introduction of IoT devices, with the increased amount of data that needed to be collected and processed, boosted this process. Grid, cloud, fog and, lately, edge computing constitute the path of "computing" paradigms of this evolution process. These technologies progressively moved the (part of) data processing load from a centralized system to a fully distributed one that is "closer" (and may consist of part of) the IoT devices. In parallel, there have been a number of proposals for intelligent networking, whose aims spans from reducing the burden of network management like in Software defined networks (SDN) [43], e.g., by separating the "control" layer from the application one, to ease the access content like in Content Centric Networking (CCN) [44], e.g., by defining protocols to that allow naming of objects on the network as in Named Networks (DND) [45], by abstracting the concept of "address" associated to each object.

While modern distributed computing paradigm constitute enabling technologies for the development of services that were impossible before, from the point of view of software distribution, a key role is played by future internet technologies. In this respect, such technologies allow to base the availability and scalability of the software distribution system, on availability and scalability properties guaranteed by the underlying intelligent networking system. Furthermore, in some case, the primitives providing by the networking system, impact on the ease of design and implementation on the update service.

### 3.6. Blockchains and DLTs

Since their introduction in 2008, blockchains have been used in different contexts to fulfil different objectives. Briefly speaking, a blockchain is a sequence of blocks, each

containing multiple transactions, that are linked together in a linear and immutable way. Immutability has to be intended in the sense that, once a transaction is registered in the blockchain, it cannot be removed or its "position" in the chain cannot be altered. It is always possible to add new blocks to the blockchain by collecting a set of transactions and binding all of them to previous blocks in the chain by means of cryptographic hash functions. All transactions stored in the blockchain are publicly accessible and verifiable. Stated in this way, blockchain appears to be a special kind of distributed database. For example, a node might compare the checksum of the software it's going to install with the corresponding one stored in the blockchain. The interest in blockchains is due to the fact that they are implemented by means of distributed ledger technologies (DLTs), that is a set of technologies that allow the implementation of a geographically distributed blockchain, with the guarantee that replicated data are consistent. Security of such technologies comes from the security of the cryptographic primitives used to construct the blockchain and by the fact that corrupting a sufficient number of nodes while they're reaching the consensus on the next node to add to the chain, is considered impossible.

Modern blockchains offer the possibility of interacting with external applications by means of smart contracts, that are procedures that are stored on the blockchain, executed by the blockchain using inputs coming from external applications and data stored on the blockchain. The authenticity and integrity of such procedures is inherently guaranteed by immutability of information stored on the blockchain. For example, it might be possible to write smart contracts that verify complex dependency properties during the installation of multiple (authentic) software packages. This interaction distributes the load of software verification among the nodes on the blockchain, while guaranteeing the security of the results.

## 4. Software Distribution Systems

One of the problems related to firmware update is the setup of an infrastructure that allows secure and reliable software distribution. The need of this infrastructure arose as soon as the software industry started using the Internet to distribute software and its updates. With this regards, major software distribution infrastructures use classical cryptographic tools, namely, hash functions and public key cryptography, to guarantee software security.

However, as stated in the previous sections, such solutions are not applicable to software/firmware updates for IoT/embedded systems, essentially for two reasons. The first one is the limited resources, both in terms of bandwidth and computational capabilities, of devices that may not be able to execute costly cryptographic primitives. The second one is the number of devices that should be updatable at the same time.

### 4.1. Secure Software Distribution Models

Given the widespread of IoT devices, there have been different attempts towards a standardization of the firmware update process. The IETF Software Updates for Internet of Things (SUIT) group released the RFC [36] in which basic requirements for the distribution infrastructure are presented.

One interesting issue pointed out in [36] is the definition of a standard software manifest, a document that describes the software along with its required dependencies. However, from a technical point of view, this RFC does not discuss installation robustness, timeliness of delivery, energy-efficiency of update procedures.

The SUIT group identified six stakeholders that are involved in the firmware update.

- The device. The key element in this model is the device on which the firmware has to be installed/updated.
- The firmware Author, who creates the image to be installed;
- Device Operator runs the day-by-day operations of the fleet of IoTs;
- Network Operator is responsible for the operations of the network to which IoT devices are connected;

- Trusted Provisioning Authority (TPA), the entity who is responsible for defining trust anchors and update policies for the update process;
- User who actually uses the IoT fleet via web or other devices.

This is the minimal set of stakeholders that can be found in an IoT deployment. However, there are cases in which some of these stakeholders may collapse to a single entity, e.g., network and device operators might be the same entity. Similarly, there are actual deployment in which the same stakeholder may appear in multiple instances, each with different privileges, e.g., multiple users in the same network or multiple networks managing the same fleet of devices.

The RFC also provides a minimal set of functionalities that need to be provided in order to guarantee the security of firmware update. These functionalities are:

- The status tracker, consisting of a server component and a client component, that is responsible for identifying the availability/triggering the update process and convoying information about the hardware and the available firmware.
- The firmware consumer, that receives the firmware and the manifest, interacts with the status tracker and executes the firmware update.
- The firmware server that distributes the firmware images and their corresponding manifests.
- The bootloader that is executed when the devices starts up and, if deemed appropriate, executes the newly installed firmware or rolls back to the previously installed one.

Most of modern IoT devices can execute the firmware update autonomously, without the need of being cable-connected to an external driver. This means that, besides having the capability of using the network to connect to the firmware server, they need to have onboard elements like the manifest parser, the capability of writing to a persistent storage, the ability to post-process the received image, e.g., decrypting/decompressing the image or verifying its integrity/authenticity.

Firmware updates may be either client-initiated, via regular polling executed by status tracker onboard the IoT device, or server-initiated, through in a push-like procedure where the (server) status tracker selectively informs the IoT of the need/existence of a new available firmware release.

The RFC discusses multiple possibilities that allow to transfer the firmware from the author to the device. Furthermore, different possibilities are available for installing the new firmware. Every possible decision on the specific selection of the transfer and installation procedures strongly depend on the actual application scenarios and it requirements. Clearly, firmware installation is security sensitive procedure that, in most cases, requires the interaction of the firmware consumer with the bootloader that restarts the device and executes the last steps of the installation procedure.

### 4.2. Threat Model

The threat model defines the capabilities an adversary may put in place when attacking the system, indented as the set of operations the adversary may execute along with the type of information that are available for the attack. Clearly, systems that may be secure in one model may become unrecoverably insecure in a different one. It is thus crucial to precisely define the threat model at the very beginning of the system design.

When dealing with software/firmware updates in IoT settings, there might be different features that may influence the threat model, e.g., attended vs unattended devices, high speed vs low speed connections, existence of cryptographic hardware etc. However, the following basic features have to be defined in every system:

- Passive vs. active adversaries. An adversary is passive if he can only eavesdrop part or all the communications among parties. An active adversary, instead, may try to interfere with the correct execution, e.g., by injecting messages.
- External adversary. This type of adversary does not have access to any device on the network. Nevertheless, it tries to attack the system by mounting attacks tries to induce

the installation of non-legit software images, impose the external code-injection/denial or service/rollback attack.

- Internal adversary-malicious/fraudulent consumer. The adversary owns/has access to one or multiple devices involved in the software upload process. Under this hypothesis, the adversary may try to obtain/induce the installation of some software his device(s) is not entitled to, e.g., in case of license infringement, installation of non-paid features or maliciously inducing device/service malfunctioning.

A typical threat model in the area of secure firmware/software update considers external and active adversaries, trying to induce the installation of wrong/modified versions of the software or to induce the rollback to previous versions of the software, e.g., in order to exploit some known security flaw.

### 4.3. Security Requirements

A key element in the design and deployment of a secure firmware update system is the clear definition of the security properties that the software update system has to guarantee, along with the corresponding deployment restrictions. Obviously, an insecure software/firmware update system may represent a security threat that is more dangerous than the one induced by the software bug it's trying to fix. Classical security properties can be instantiated in the context of software updates as follows:

Software authentication guarantees that the device is able to identify the author of the software and the manifest before its installation. This property clearly prevents the installation of software produced by malicious third parties.

Software integrity guarantees that the manifest or the software that is going to be installed has not been modified by any third party. Notice that, whenever the software consists of multiple modules/packages, software integrity guarantees that all modules/packages have been correctly delivered.

Confidentiality protection might be required in different contexts. This property guarantees that the software and its manifest can be read only by the device that is intended to install it. The need of protecting the software from unauthorised parties may be due to the fact that the software is intendeLd as an intellectual property of its authors. Another reason for which such property may be requires is the fact that an adversary may reverse-engineer the software in order to find bugs or vulnerabilities. In [36], confidentiality protection is considered as an optional property.

In order to guarantee "classical" security properties in IoT devices, as we have seen in Section 3.4, there exist plenty of cryptographic primitives that have been specifically designed to work on constrained devices. It is still necessary to continuously monitor advances in the cryptanalysis of these primitives and to properly choose the key size for such primitives that do allow to select such security parameter. As usual, the usage of a secure primitive can turn into a completely insecure deployment whenever the operational context is not considered properly. This means that the primitive has to be considered as a building block of a more complex protocol that involves all parties listed in Section 4.1 and whose security should be considered as a whole.

### 4.4. Challenges and Solutions

Besides the above classical security properties defined in the previous section, there are a number of properties that arose in the last years and that have been set out and dealt with in different papers.

It has been shown, e.g., [5,46–48] that, using class-4 devices, as defined in [5], i.e., devices with few KBs of RAM, with 8-bit processors, that do support ECC asymmetric cryptographic primitives, in conjunction with class-1 or class-2 devices, to secure the system operations and, in particular, the software update process. It is thus crucial to have one or more "PC-like" devices in loop in order to be able to execute the high level computations and to work as local decision point for the installation policy. At the same time, securing deployments with class-5 devices is still possible but requires ad-hoc solutions that need

use/implement symmetric cryptographic primitives, while using techniques like preloaded keys to secure and authenticate communications.

The ASSURED Architecture [49] performs Secure Software Updates for embedded devices with particular attention to the "IoT ecosystem". In order to cover different classes of target devices, the design of ASSURED offers different solutions for inter-parties authentication, secure communication and remote attestation. In particular, ASSURED provides end-to-end security and local update authorization leveraging TMF's TA management facilities as long as the pool of devices to be updated includes a TMF-enabled controller.

In some real-world scenarios, especially in the context of IoT, devices are unattended, and/or deployed in inaccessible locations and/or can connect only wirelessly to the update server [48–51]. In such scenarios, the security of the update procedure should consider the need of using an over the air connection that is, by itself, insecure. Because of this, confidentiality and authentication might be ensured in an end-to-end fashion where the destination-end is the device and, depending on the actual deployment scenario, the source-end might vary from the author's distribution server to network operator one.

Typical IoT deployments consists of thousands of devices that might, potentially, require concurrent updates. From the point of view of device manufacturers, there is the need of setting up an update infrastructure that should be able to potentially update millions of devices at the same time. From this discussion, it is clear that scalability is a feature that need to needs to be provided by software update infrastructures [52–54]. Furthermore, as observed in [49], update infrastructures cannot require interactive procedures and should minimize the work required on the device side. Another issue to be solve, in such scenarios, is to properly identify which device is authorized/required to download and install a specific piece of software. In [55] the authors take advantage of naming primitives provided by Named Networks [45] and attributed based encryption schemes, by assigning each image a name that depends on the sequence of attributes that characterize the device for which such image has been built. In this way, each device have a specific set of "attributes" will be able to retrive the encrypted image it needs and to decrypt it using its own attributes.

In many cases, devices are running applications by different authors on top of the firmware produced by the manufacturer [56–60] A observed in [60], different types of software might have different degrees of criticality. Clearly, the firmware running on a device is the most critical software whose insecurity threatens the operations of the device. On the other hand, an insecure payload running on the device is less critical in the sense that it might lead to the unavailability only on the service it executes, while the other services running on the same device might be completely unaffected. One key issue in such multi-application/multi-tenant devices is the capability to enforce the system stability, i.e., the capability to prevent that an inappropriate update badly interfere with the firmware or other applications on the device. This property, already identified in [61] in the enterprise-wide applications, considered the hidden dependencies, i.e., the ones induces by the complexity of the "enterprise" system. Authors considered different faults types, all implicitly assumed to be part of accidental (non-malicious) errors. There were different solutions for this model, e.g., upgrading one node at the time or enforcing and verifying atomic updates in a sealed compartment before switching to the new software version. However, these solutions implicitly assumed the capability of detecting a fault in a compartment or the even stronger assumption of a centralized supervision of the update process. However, in the context of IoT/mobile devices, an active adversary might try to break software dependencies by installing inappropriate software [59,62], e.g., trying to break licensing rules or to actively operate to make the system unstable.

Recently some authors consider the convergence of IoT deployments with the world of cloud/fog storage and computing. In such scenarios, the dynamic nature of the network topology on the cloud side is guaranteed by the SDN paradigm [54]. Although security is clearly a prerequisite in such deployments, interoperability among different devices for different computing services and/or storage services, e.g., smart home [63], smart cities [64], mobile storage [65], etc.

Security issues related to power draining attacks have always been central in the research on battery operated devices. In particular, maliciously modified firmware may attach directly the battery management system [66]. In these cases, different techniques are available, ranging from preventing the attack by monitoring of the device power consumption [67,68] to recovering from a modified firmware using the support of a blockchain [66].

A further step to automate and enforce IoT firmware/software updates in heterogeneous deployments is the one of using blockchains [50,69]. As stated in the previous section, corrupting data stored in a blockchain is impossible. Furthermore, the geographic distributed nature of blockchains is a guarantee for scalability for the resource demanding task of software distribution. Notice that this solutions add new components in the model defined in Section 4.1. Indeed, the blockchain becomes a new component that lays between the manufacturer and the site/network operator and is used by the latter to identify and authenticate software/firmware components before their installation on the final devices.

There are solutions, e.g., [60], that go in the opposite direction of imposing the direct operator control over software/firmware requests. Typically, such solutions assume the existence of a human operator who is responsible for identifying devices and software that, in the domain under her control, need to be updated. Furthermore, she has the authority to impose the installation of software to all the devices. As usual, whenever there is a human in the loop, the classical problems of continuous user authentication [57,58], usability of authentication devices [70,71] have to be carefully considered.

In Tables 1 and 2, we summarize some of the results reported above.

**Table 1.** Comparing existing solutions.

|  | **Zandberg et al. [46]** | **De Sousa et al. [48]** | **Asokan et al. [49]** | **Karthik et al. [72]** |
|---|---|---|---|---|
| Required Computational Capabilities | Class-2 devices and lightweight cryptography | Class-3 devices with TCP-IP capabilities | Heterogeneous devices assigned to a primary domain controller with TEE + TMF | Heterogeneous devices grouped with a primary domain controller with PC-like capabilities |
| Unattended operations | Initial update module + preloaded trust anchor | Bootloader rolls back invalid updates | Remote attestation to check updates | N/A |
| End-to-end security | Signed Manifest and firmware | Device authentication via preloaded credentials | Authorization tokens in update packages | Package source authentication via digital signature |
| Multiple stakeholders | Legacy RTOS + single firmware image | N/A | TEE + secure boot to protect code and keys | N/A |
| Scalability | Multicast protocol suite | Requires interactive protocol | Package delivery outsourced to a CDN | Requires interactive protocol |
| Heterogeneous devices | Standard OS, libraries, runtime and protocol stack | N/A | Not Specified | Not Specified |
| Network Intelligence | N/A | Device management in cloud | Payload delivery through a CDN | N/A |

**Table 2.** Comparing existing solutions.

| | Anastasiou et al. [50] | Ambrosin et al. [55] | Seitz et al. [60] | Catuogno et al. [59] |
|---|---|---|---|---|
| Required Computational Capabilities | LoRA Alliance complaint devices | Lightweight cryptography | PC- like controller in the loop + lightweight cryptography | TEE |
| Unattended operations | Public keys preloaded in devices | Public keys preloaded in devices | Software updates are partially automated (Human in the loop) | Software update may be fully automatic |
| End-to-end security | Software packages signed at source. Integrity and installation policy driven via smart contracts | Encrypted and signed payloads using ABE | Software packages signed at source | Software packages signed at source + installation policy enforcement via secret sharing |
| Multiple stakeholders | N/A | Not specified | Legacy OS + Multiple third party applications | Legacy OS + Multiple third party applications |
| Scalability | Multicast protocol suite + geographically distributed blockchain | Use of Named Networks | N/A | Not specified |
| Heterogeneous devices | LoRA Alliance complaint devices | Not specified | Not Specified | Not Specified |
| Network Intelligence | Underlying Blockchain | Named Networks | N/A | N/A |

### 4.5. Challenging Issues

The literature on the IoT devices is vast due to the enormous number of possible variables that influence each deployment. Just to name few, device heterogeneity in multi-manufacturer/multi-tenant deployments. In the specific context of software/firmware update, the need of allowing the definition and enforcement of local installation policies that might contrast with the manufacturer defined ones.

There are some issues we believe require specific attention when deploying a software update system in the IoT/embedded operational context.

The first one is the need of revoking and renewal of corrupted keys. This issue is particularly relevant in all deployments in which devices are unattended. However, it is gaining increasing importance also in cases in which devices are somehow secure, e.g., smart home devices, because a corrupted key provides access to an adversary to an environment that was allegedly assumed secure. Key renewal/revocation might, in principle involve a specific device. However, if the key management system is poorly designed, it might induce the need of involving all devices in a given domain.

The possibility of giving control to device owners is another crucial point. If we think, again, to the case of smart home, the owner has all the rights to decide which device has to be updated. Unfortunately, users typically do not have the necessary expertise to take informed decisions. In a heterogeneous single-manufacturer environment, this is not really an issue. The complexity of decision making becomes high in multi-manufacturers deployments in which a properly designed middleware might provide a good support to the device owner.

The use of blockchains as support for software/firmware update infrastructures might have an impact of scalability and security. However, there are some issues that still need to be properly considered in an actual deployment. Blockchain transactions typically have a cost that is used to support providers of the DLT on which the blockchain runs. In some cases, such costs can be extremely high. A second important issue is the number of transactions per second the blockchain can provide. In many cases, despite the interest in such technologies, the currently available solutions in many cases cannot provide a sufficient number of transactions per second that might be of interest for the purposes of

software updates of worldwide distributed IoT devices. To this aim, a possible candidate might be the Algorand blockchain [73] that has extremely low costs, currently $0.0002 per transaction, and high number of transactions per second, currently up to 6000 TPS.

## 5. Conclusions

In this paper we have analyzed the problem of software/firmware updates in the context of IoT/embedded devices. We have argued that complexity of the IoT world is essentially related to heterogeneity and multi-manufacturer deployments. Furthermore, the need of executing resource demanding security primitives on resource constrained devices, makes the design and deployment of secure distribution infrastructures a challenging task. We have discussed a set of enabling technologies and methodologies that are currently used to deploy secure systems. We have also discussed some challenges that naturally appear in the this specific context and listed some solutions proposed in the literature. We finally presented some issues that we believe need to be carefully considered when designing and deploying a new software update infrastructure in a specific application scenario.

**Funding:** This work was partially supported by project SERICS (PE00000014) under the MUR National Recovery and Resilience Plan funded by the European Union—NextGenerationEU.

**Conflicts of Interest:** The authors declare no conflict of interest.

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
