# Peer review of "Secure Firmware Update: Challenges and Solutions"

_cryptography, doi:10.3390/cryptography7020030_

Round 1

Reviewer 1 Report

The paper reviews a quantity of aspects in IoT firmware update security. Though it contains very little novelty (being it a review paper), it is very well written and pleasing to read.

I think the major problem of the paper is that authors neglected the emerging aspect of post-quantum security in firmware updates. The RFC 9019 itself (that the authors cited in [38]) says: "[...] since signature schemes based on RSA and Elliptic Curve Cryptography (ECC) may become vulnerable to quantum-accelerated key extraction in the future, unchangeable bootloader code in ROM is recommended to use post-quantum secure signature schemes such as hash-based signatures . A bootloader author must carefully consider the service lifetime of their product and the time horizon for quantum-accelerated key extraction. At the time of writing, the worst-case estimate for the time horizon to key extraction with quantum acceleration is approximately 2030, based on current research."

I think the authors should discuss this important aspect, and cite some recent literature about it for example:

* Banegas, G., Zandberg, K., Baccelli, E., Herrmann, A., & Smith, B. (2022, June). Quantum-resistant software update security on low-power networked embedded devices. In Applied Cryptography and Network Security: 20th International Conference, ACNS 2022, Rome, Italy, June 20–23, 2022, Proceedings (pp. 872-891). Cham: Springer International Publishing.

* M. L. Manna, P. Perazzo, L. Treccozzi and G. Dini, "Assessing the Cost of Quantum Security for Automotive Over-The-Air Updates," 2021 IEEE Symposium on Computers and Communications (ISCC), Athens, Greece, 2021, pp. 1-6, doi: 10.1109/ISCC53001.2021.9631426.

Typos/minor comments:

- is a family fo authenticated [Page 6]

- the entity may posses [Page 6]

- ABE schemes -> "ABE" acronym has never been introduced.

- Some authors consider te Notice that this [Page 11]

- softwares -> "software" is uncountable. please use "pieces of software" instead [Page 11]

Author Response

We thank the reviewers for their comments and suggestions. 

We have added a paragraph in Section 3.4 Lightweight cryptography, describing post-quantum systems and reported initiatives in the field. 

Reviewer 2 Report

In this paper, a review of current limitations posed to software distribution systems is introduced for embedded/IoT devices. The challenges faced the researchers in this area have been identifying and the authors proposed the corresponding solutions. However, there are some major comments should be addressed.

1.     The manuscript should be considered as a review paper and not as article.

2.     The authors should review a number of current published papers and surveys on the field such as “Secure firmware Over-The-Air updates for IoT: Survey, challenges, and discussions”, 2022-Elsevier           

3.     The manuscript needs some tables and figures that summarize the current state-of-the art methods and approaches for readability

4.   The manuscript needs some statistical analysis or some criteria on benchmarking, data, and models and also including the authors’ findings.

 Moderate editing of English language

Author Response

We thank the reviewer for the comments and suggestions.

We will ask the editor to modify the paper type to review. 

We have considered the other comment and added a table summarising the analysis of some papers in the literature. 

We think it is difficult to evaluate papers in this context using statistical analysis since solutions are thought to work in different contexts and, thus, are incomparable using this type of analysis.

We have added two tables in which we compare some of the papers in the literature, explicitly listing some comparison criteria.

Reviewer 3 Report

This paper shows Secure firmware update: Challenges and solutions. They review the current limitations posed to software distribution systems for embedded/IoT devices, consider challenges that the researchers in this area have been identifying and propose the corresponding solutions.

1) The background to Secure firmware update should be clearly shown. You had better to give Advantages and disadvantages of existing methods, why or form which view you start your research point. For software and hardware or IoT device, maybe there are different requests for the Secure features.

2) In introduction part, you have to reconstruct it according to the sequence of background, problem to state,existing methods with disadvantage. The section of related works should be sorted based on different method or methods. For Secure,there are many research, such as anomaly detection DOI:10.1109/TNSE.2022.3163144, IoT Secure DOI:10.1007/s11036-021-01846-x and so on.

3) and you can improve English

4) the references can be improved. I suggest the author to give tables and figures for comparing the related works. It can help the reader to understand the progress of the secure firmware update

can improve English

Author Response

We thank the reviewer for the comments and suggestions. 

We have expanded  the introduction so to better explain the application context. We think we might be misleading to provide in the introduction pros and cons of different solutions since they've been designed for different contexts and, thus, are incomparable.

 Following the reviewer's comments, we have inserted two tables in which we compare some of the systems in the literature and we've reviewed the English in the paper. 

Round 2

Reviewer 1 Report

The authors addressed my concern. Only one typo: "the algorithsm".

Author Response

We've spellchecked the paper to correct all typos. 

Thanks

Reviewer 2 Report

Thanks to authors for updating the manuscript according to the comments. However, the type of the manuscript is still marked as "Article". So, please change the type as "Review".

Best regards

Minor editing of English language required.

Author Response

We've spellchecked the paper. 

We will ask the editor to change the article type.

Thanks

Reviewer 3 Report

I checked this paper and would like to vote to accept this paper.

It can be improved before submitting final paper.

Author Response

We've checked the spelling and the English.

Thanks.

Round 3

Reviewer 2 Report

Thank you for addressing the comments.